# One-Step Synthesis of Silver Nanowires with Ultra-Long Length and Thin Diameter to Make Flexible Transparent Conductive Films

**DOI:** 10.3390/ma12030401

**Published:** 2019-01-28

**Authors:** Yuxiu Li, Ximin Yuan, Hongwei Yang, Yunxiu Chao, Shuailong Guo, Chuan Wang

**Affiliations:** State Key Laboratory of Advanced Technologies for Comprehensive Utilization of Platinum Metals, Kunming Institute of Precious Metals, Kunming 650106, China; liyuxiu@ipm.com.cn (Y.L.); yxm@ipm.com.cn (X.Y.); cyx@ipm.com.cn (Y.C.); gsl@ipm.com.cn (S.G.); wangxiaochuan@ipm.com.cn (C.W.)

**Keywords:** Silver nanowires, Solvothermal, Transparent conductive film, Sheet resistance

## Abstract

High aspect ratio silver nanowires (AgNWs) with ultra-long length and thin diameter were synthesized through bromine ion (Br^−^)-assisted one-step synthesis method. The bromine ions were used as pivotal passivating agent. When the molar ratio of Br^−^/Cl^−^ was 1:4, the average diameter of AgNWs was as low as ~40 nm, the average length was as high as ~120 μm, and the aspect ratio reached 2500. Networks of AgNWs were fabricated using as-prepared high-quality AgNWs as conducting material and hydroxyethyl cellulose (HEC) as the adhesive polymer. As a result, a low sheet resistance down to ~3.5 Ω sq^−1^ was achieved with a concomitant transmittance of 88.20% and a haze of 4.12%. The ultra-low sheet resistance of conductive film was attributed to the long and thin AgNWs being able to form a more effective network. The adhesion of the AgNWs to the substrate was 0/5B (ISO/ASTM). The insights given in this paper provide the key guidelines for bromine ion-assisted synthesis of long and thin AgNWs, and further designing low-resistance AgNW-based conductive film for optoelectronic devices.

## 1. Introduction

With the emergence of flexible photoelectric devices and the scarcity of indium resources, great efforts have been made to develop new flexible transparent conductive films (TCFs, an essential element of flexible optoelectronic devices) to replace traditional indium tin oxide (ITO). Until now, conductive polymers [1,2,3], silver nanowires (AgNWs) [4,5,6], metal grids [7,8], graphene [9,10,11,12], carbon nanotubes (CNTs) [13,14,15], and other random networks of metallic nanowires (NWs) [16,17] have been regarded as emerging candidates of ITO. Among these materials, carbon-based nanomaterials including CNTs and graphene exhibit high sheet resistance and low optical transparency due to either low intrinsic carrier concentration or high tube-tube resistance. Many researchers have regarded AgNWs and metal grids as preferred alternative [4,8], but the metal grids will cause serious moiré interference phenomenon, and its production process is more expensive [18,19]. Compared to metal grids, AgNWs have high mechanical flexibility, excellent electrical conductivity and optical transparency [20,21]. Its smaller line width will not cause moiré interference phenomenon. Meanwhile, AgNW-based TCFs have a small bending radius, which will lead to a small resistance change rate, and thus offer many advantages in devices of curved display.

To our knowledge, the properties of AgNW-based TCFs mainly depend on characteristics including the nanowire structure and the overall network morphology. In general, the length, diameter, and aspect ratio of nanowires are critical factors for enabling the high transparency with a low haze, low sheet resistance, and superior mechanical compliance and strength [22]. However, they are usually hard to achieve simultaneously. Kim et al. claimed that a reaction time of 4 h can synthesize AgNWs with a diameter of 62.5 nm, but only about 13.5 μm in length [23]. Atwa et al. experimentally showed that a high temperature of 170 °C can grow nanowires, however the length is below 15 μm with diameter over 35 nm [24]. Jiu et al. used a simple polyol process to grow very-long AgNWs by only adjusting the stirring speeds. The length is 80 μm with a diameter in the range of 75–85 nm [22]. Thus, the search for synthetic route capable of controlling the length and diameter of AgNWs is still ongoing.

Through performing an extensive literature search and seriously comparing and summarizing the factors that affect the morphology of AgNWs in the synthetic process, it was found that, except for the reaction temperature and time, it appears that the varieties and quantities of nucleant are crucial. Search results showed that bromide ions (Br^−^) as co-nucleant can effectively reduce the diameter of AgNWs, but a small AgNW length is always obtained owing to the harsh conditions. Lee et al. reported that employing NaCl and KBr as nucleants could control the diameter of AgNWs in the range of 15–30 nm, however the length was below 20 μm [25]. Zhang et al. claimed that KBr was beneficial for decreasing AgNWs diameter but unable to generate nanowires effectively alone, thus they used NaCl, FeCl_3_ and KBr as co-nucleants and achieved thin AgNWs with an average diameter of 26 nm, but only about 21 μm in length [26]. Wan et al. synthesized AgNWs with an average diameter close to 20 nm by a polyol reduction method in the presence of NaBr and NaCl, but the nanowire length approached 30 μm [27]. To the best of our knowledge, only few studies have reported that Br^-^ decreased diameter of AgNWs while increasing length, and through controlling the growth of AgNWs obtained AgNWs with a large aspect ratio [28].

In this study, we showed that the diameter of AgNWs in a solvothermal process can be decreased by adding potassium bromide, and by adjusting the molar ratio of KBr/NaCl the length of AgNWs can be increased. AgNWs with diameter of ~40 nm, length of ~120 μm, and aspect ratio up to 2500 were obtained by one-step synthesis. The as-synthesized high-quality AgNWs were transferred to hydroxyethyl cellulose (HEC) to form conductive ink and coated onto polyethylene glycol terephthalate (PET) to fabricate AgNW-based TCFs with a transmittance of 88.20% and a haze of 4.12% while achieving a low sheet resistance of ~3.5 Ω sq^−1^. The possible reason ultra-long AgNWs achieve lower resistance is also discussed.

## 2. Materials and Methods 

### 2.1. Materials

All chemicals were of analytical grade and used as received without further purification. Silver nitrate (AgNO_3_, ≥99.8%), sodium chloride (NaCl, ≥99.5%) and potassium bromide (KBr, ≥99.0%) were purchased from Sinopharm Chemical Reagent Co., Ltd. (Shanghai, China) poly-vinylpyrrolidone (PVP, Mw ~1.3 × 10^6^) was acquired from Sigma Aldrich (Shanghai, China). Hydroxyethyl cellulose (HEC) was purchased from Dow (Shanghai, China). Ethylene glycol (EG, ≥99.0%) and ethanol (C_2_H_5_OH, ≥99.0%) were obtained from Xilong Scientific Co., Ltd. (Shantou, China). Polyethylene glycol terephthalate (PET) was gained from Yingshang Electronic Materials Co., Ltd. (Shenzhen, China). Deionized (DI) water was purified by laboratory water purification system with a resistivity of 18.2 MΩ·cm and a working temperature of 25 °C.

### 2.2. Preparation of Thin and Long AgNWs

As the most common reagent, KBr was chosen as the chemical auxiliary of synthesize ultra-long AgNWs. As is known, the existence of KBr will lead to uncontrollable nanoparticles. To avoid the formation of many nanoparticles, and prepare ultra-long AgNWs, while further performing the function of KBr thin AgNWs, NaCl was used to assist KBr, and collectively realize the synthesis of ultra-long AgNWs. In a typical synthetic experiment, 1:4 molar ratio of Br^−^/Cl^−^ was dissolved into 160 mL EG, as “a” solution (the molar ratio of 1:4 was proved to be optimal for the length and diameter of AgNWs in our incipient experiments). Then, 1.792 g of AgNO_3_ and 11.12 g of PVP were dissolved into 240 mL EG, to get transparent solution, as “b” solution. Later, the “b” solution was poured into “a” solution to form a light orange mixed solution. After magnetically stirring for 10 min, 400 mL of the mixture was transferred into a Teflon-lined stainless steel autoclave with a capacity of 500 mL and reacted under solvothermal conditions at 170 °C for 2.5 h. The autoclave was cooled down to room temperature in a standard atmosphere. The grayish-green product was centrifuged, and alternately washed with ethanol and DI water, dispersed in a certain amount of DI water. The solid content of AgNWs was measured to be 0.95%.

### 2.3. Preparation of AgNW Ink

AgNW ink was prepared by a simple physical method. Specifically, AgNWs were dispersed into the solvent system that matched matrix resin. First, 1.0 g HEC was dissolved into 100 mL DI water to form a 1.0 wt % solution. Then, 12 mL of above solution were transferred into a beaker, and stirred at room temperature. After that, 26 mL of aqueous AgNWs dispersion liquid were immersed into the solution, and 12 mL of DI water was further added, making the solid contents of AgNWs and HEC 0.5% and 0.25%, respectively. After being stirred for about 3 h, the final ink was obtained for fabricating TCFs.

### 2.4. Fabrication of AgNW-Based TCFs

The AgNW-based TCFs were fabricated on a BEVS1811/2 bar coater with a vacuum plant (Zhuhai Tianchuang Instrument Co., Ltd., Zhuhai, China). The fabrication process comprised the following steps: first, the bare PET substrate was adsorbed onto the platform by vacuum. Then, a bar of 20 μm was put down to hold the substrate, and dumped the AgNW ink on substrate. The length and speed of coating were set as 10 cm and 300 mm/s, respectively, and then the start button was pressed. The bar pushed the ink to glide over the substrate. After that, the coated-film was taken down and put in an air dry oven of 130 °C and curing 5 min, and obtained a flexible transparent conductive film.

### 2.5. Characterization

X-ray diffraction (XRD, Rigaku D/MAX-3B powder diffractometer, Shanghai, China) with a copper Kα radiation (λ = 1.54056 Å) was used for the crystal structure and the phase identification, where the diffracted X-ray intensities were recorded as a function of 2*θ*, and scanned in steps of 0.02° (2*θ*) from 20° to 90°. The microstructures of the sample were observed on a FEI-Versa3D field emission scanning electron microscopy (FE-SEM, FEI, Hillsboro, Oregon, USA) operating at 30 KV. Before FE-SEM test, the sample was dispersed in ethanol and further dripped on silicon wafer. The transmission electron micrography (TEM, Tecnai G2-TF30, Beijing, China) was performed on a Zeiss EM 912 Ω instrument at an acceleration voltage of 120 KV, while high-resolution transmission electron microscopy (HRTEM) and fast Fourier transform (FFT) were examined on a Philips CM200-FEG microscope (200 KV, *Cs* = 1.35 mm, Tecnai G2-TF30, Beijing, China). Due to the very brittle structure of AgNWs, the sample used for TEM was dispersed in ethanol with no need for ultrasound treatment, and the dispersion was then dropped on carbon–copper grid. The UV-vis spectra of the sample were performed on a UV-vis spectrophotometer (PERSEE Genera TU-1901, Beijing General Analysis Instrument Co., Ltd., Beijing, China) with a scan range from 190 nm to 900 nm. SGW-820 transmittance and haze analyzer (Shanghai Yidian Physical Optical Instrument Co., Ltd., Shanghai, China) were employed to study the transmittance and haze of AgNW-based TCFs, and a bare PET film was used as blank test. The four-point probe (SB100A/2) was used to examine the sheet resistance of AgNW-based TCFs with a probe current of 2 mA. Tests for adhesion were conducted by a Cross-Cut Tester scratching (Aipli, Shanghai, China) the PET substrate with AgNW-based TCF. Then kapton tape was stuck to the film, and then peeled off. The pattern obtained was examined visually and classified by means of a reference table. According to the standard applied, an identification number was assigned to classify the adhesion properties of the tested coating. The grade of 0/5B indicates the best adhesion.

## 3. Results and Discussion

The XRD pattern of the as-synthesized sample is depicted in Figure 1. In terms of XRD pattern, it can be fully indexed to a pure phase of face-centered cubic (fcc) Ag crystal (a = b = c = 4.086 Å, space group Fm-3m (225)) by comparing the curve with JCPDS (99-0094) of Ag. The obvious five peaks at 38.11°, 44.30°, 64.44°, 77.40°, and 81.54° corresponded to (111), (200), (220), (311), and (222) Bragg reflections of Ag. No traces of other compounds such as Ag_2_O could be detected, indicating the high crystallinity and purity. Besides, the intensity of (111) Bragg reflection was largest, suggesting that the AgNWs grew along the (111) Bragg reflection. Its crystallite size was estimated to be about 20 nm, which was a prerequisite for the formation of thin AgNWs.

Figure 2 demonstrates the morphology of the as-synthesized AgNWs by FE-SEM. It was observed that a typical wire-like structure was obtained by solvothermal method, as shown in Figure 2a. The great mass of wire lengths was measured to be about 120 ± 15 μm using DigitalMicrograph and ImageJ software (Keyang International Trade (Shanghai) Co., Ltd., Shanghai, China). A closer examination revealed no other silver nanostructures, such as nanoparticles, nanorod, and nanocube, which indicates as-synthesized AgNWs had high purity and uniformity. The detailed and typical magnified morphology are presented in Figure 2b. It was observed that a smooth surface was formed on AgNWs. The diameter of AgNWs ranged from 24 nm up to 45 nm, and the average diameter was about 40 nm, which illustrates a high aspect ratio of ~2500. It is well known that thin and long AgNWs will greatly influence the photoelectric properties of AgNW-based TCFs [29].

The detailed structures of the obtained AgNWs were further observed via TEM, HRTEM and FFT. Figure 3a,b clearly shows the structure characteristic of wire with a thin and uniform diameter and a narrow size distribution. As demonstrated in Figure 3a, the surface of AgNWs had an obvious capping layer, corresponding to residual PVP. The PVP layer was difficult to completely eliminate, even after washing and centrifuging repeatedly, and led to loose contact between AgNWs and induced capacitive junctions, thus affecting electrical properties of AgNW-based TCFs [30]. According to the magnified TEM image (Figure 3b), the diameter of AgNWs varied from 33 to 45 nm with a narrow size distribution, which further supported the above FE-SEM results. In addition, one can see that the surface was smooth without tiny structures. The magnified TEM image gave further insight into the morphology of AgNWs, as shown in Figure 3c, where two special parts are marked by the yellow dashed line: the tip of AgNWs, clearly showing the formation of a five-fold twinned structure [31], and demonstrating the existence of a twin-plane structure parallel to the crystal growth direction. An HRTEM image at higher magnification of the same sample (Figure 3d) exhibited coincident lattice fringes. The HRTEM micrograph of the special part marked by the yellow dashed line in Figure 3d is shown in Figure 3d (2). Figure 3d (2) shows clear well-developed lattice fringes. These results together with the above XRD analysis strongly confirm the high crystallinity of our AgNWs. The FFT pattern shown in Figure 3d (1) further proved that AgNWs possess twin crystalline structure.

The evolution of morphologies from original nucleation to high aspect ratio AgNWs was demonstrated by schematic diagram, as compared in Figure 4. As temperatures rose, the Ag ions were continuously reduced to Ag atoms, multiplying twinned particles (MTPs), and Ag nanoparticles, simultaneously sticking to the surfaces of AgCl and AgBr cubes. The MTPs grew into Ag nanorods and further into AgNWs with increased heating temperature and reaction time. Of interest, in this process, the PVP macromolecules interacted more strongly with the {100} facets than with the {111} facets, ultimately causing the formation of AgNWs. Based on the above theories, the possible reasons of thin and long AgNWs were successfully synthesized were attributed to three aspects. The first one is related with the effect of KBr, which can effectively decrease the diameter of AgNWs. As reported, first, the Br^-^ can influence the size of the initial nucleus, and the AgBr cubes are much smaller than the AgCl [32]. Then, when the AgBr releases Ag^+^, the Br^−^ adsorbs on the surface of original AgNWs, and the Br^-^ can passivate the {100} facets, limit lateral growth, and further promote the formation of thinner AgNWs [29]. Next, it is likely that the high affinity to Br^−^ leads to a slower release of Ag atoms contributing also to thin and high aspect ratio AgNWs [33]. The second is ascribed to the PVP with a high molecular weight of 1.3 × 10^6^. Due to the increase in chain length, the solution would possess a high viscosity, slowing down the growth kinetics and leading to the formation of MTPs. Furthermore, the PVP with a higher molecular would strongly adsorb on Ag nanocrystal surface, leading to a confinement to the lateral growth [34]. The third is attributed to the pressure of reaction still, making the reactive solution rapidly reach supersaturation, and promoting the formation of small-size Ag seed and thin AgNWs [35]. 

In addition to FE-SEM and TEM, the UV-Vis absorption is another characterization method for analyzing AgNWs morphology. As is known, the UV-Vis spectra can be used to describe the morphology, as Ag nanostructures with different sizes and shapes display surface plasmon resonance (SPR) bands at different frequency ranges. For AgNWs, there are two plasmon adsorption resonances, corresponding to the transverse oscillation of electrons and the oscillation of electrons along the long axis [36]. Figure 5 illustrates the UV-Vis absorption spectra of AgNWs in DI water. The absorbance peak of AgNWs occurred in the wavelength range of 340–380 nm, and showed strong absorption of near-ultraviolet and short-wavelength visible light, with a main peak at 369.38 nm and a shoulder at 353.24 nm, both corresponding to typical absorption peaks of AgNWs. The absorption spectra of AgNWs showed two obvious peaks, because of the low symmetry of the pentagonal cross section. It is well known that the complexity of plasmon absorbance increases with decreasing symmetry of the cross section [37]. Significantly, the frequency of main peak below 370 nm leads to the blue-shift of absorption peaks, suggesting the diameter of as-synthesized AgNWs is small [25].

To explore potential applications, as-synthesized AgNWs was used as a conductive material to prepare conductive ink using HEC as adhesive. The inset of Figure 6a records the appearance features of as-prepared AgNWs conductive ink. As shown in the inset of Figure 6a, the conductive ink exhibited typical atrovirens, indicating a small diameter of AgNWs. It is widely accepted that the color of AgNWs darkens with decreasing diameter of AgNWs. In addition, the addition of HEC can effectively enhance the viscosity of ink, and give it excellent printability. The flexible nature of as-fabricated AgNW-based TCFs was verified. The TCF was curved by hand, as shown in Figure 6a, indicating a good flexibility for our conductive film. On closer inspection, we found that the surface of PET is highly uniform. The latex gloves behind the conductive film could be seen clearly, illustrating the film was highly transparent (the transmittance is shown in Figure 7). Because electrical conductivity is a remarkable parameter to authenticate an excellent film, the sheet resistance of as-fabricated AgNW-based TCFs was tested by four-point probe, as shown in Figure 6b. Of importance, our result illustrates an ultra-low sheet resistance of ~3.5 Ω sq^−1^, which was much less than previous reports [38,39,40,41]. To verify the adhesion of the as-fabricated AgNW-based TCFs, an adhesion test was performed by a kapton tape, and the adhesion was estimated to be 0/5B (ISO/ASTM grade).

Afterwards, the optical property of as-fabricated AgNW-based TCFs was elucidated, as shown in Figure 7. In contrast, the optical property of bare PET substrate was tested by transmittance and haze analyzer. Figure 7a displays transmittance and haze of bare PET substrate were 91.98% and 0.09%, respectively. Figure 7b represents the optical property of as-fabricated AgNW-based TCFs, and transmittance and haze were 88.20% and 4.12%, respectively. Summing up the above results, the AgNWs can be used for conductive material, and the as-synthesized AgNW-based TCFs exhibited excellent photoelectric properties.

The mechanism for the ultra-low sheet resistance was caused by ultra-long and high aspect ratio AgNWs, as schematically represented in Figure 8. In Figure 8a, we can find that long AgNWs with small diameter can form a more effective network with smaller nanowire number density by providing longer percolation paths and reducing the number of high-resistance nanowire–nanowire contacts in the film [22], thus causing low sheet resistance. However, for short AgNWs, the nanowire number of the unit area is greater than for long AgNWs. Simultaneously, the junction number between nanowire and nanowire unit area also increases, as shown in Figure 8b. The loose connection of the AgNWs at the junction leads to a higher sheet resistance. Therefore, the optoelectrical properties of AgNWs networks depend heavily on nanowire length and diameter [42,43,44]. 

## 4. Conclusions

In summary, ultra-long and thin AgNWs were one-step synthesized by adding KBr as co-nucleant with a molar ratio of 1:4 of KBr/NaCl. Subsequently, AgNO_3_ was translated into AgBr and NaCl, and was reduced into Ag via solvothermal at 170 °C. It was found that the as-synthesized AgNWs exhibited a diameter of ~40 nm, a length of ~120 μm, and aspect ratio up to 2500. The flexible transparent conductive film based on as-synthesized AgNWs exhibited ultra-low sheet resistance of ~3.5 Ω sq^−1^, as well as high transmittance and low haze, strong adhesion, which were mainly attributed to long AgNWs with small diameter being able to form a more effective network. These findings not only offer a novel route to synthesize high-quality AgNWs, but also provide a promising method to prepare environmentally friendly aqueous AgNW-based conductive ink, and further obtain high-performance flexible AgNW-based TCFs.

## Figures and Tables

**Figure 1 materials-12-00401-f001:**
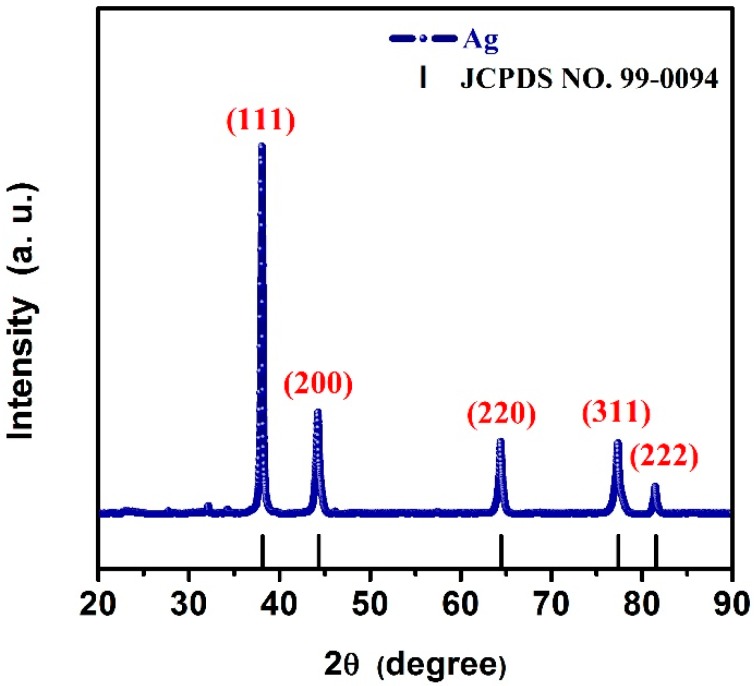
XRD patterns of the as-synthesized pure AgNWs, JCPDS: 99-0094 of the Ag phase.

**Figure 2 materials-12-00401-f002:**
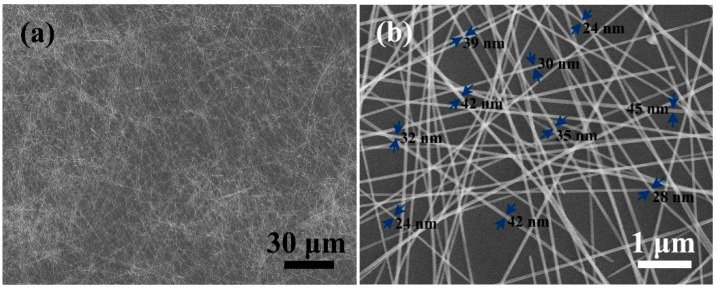
(**a**) Low-magnification FE-SEM image of the synthesized AgNWs; and (**b**) representative high-magnification FE-SEM image of the synthesized AgNWs.

**Figure 3 materials-12-00401-f003:**
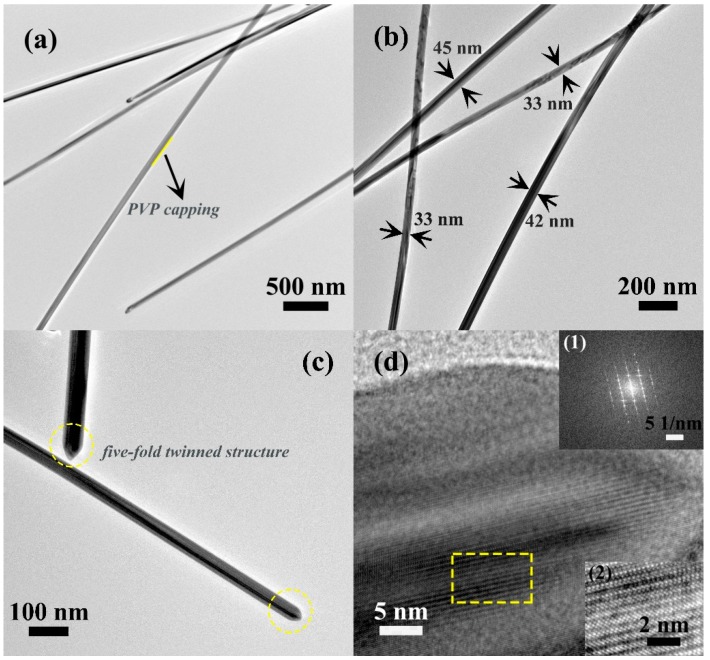
(**a**–**c**) TEM micrographs of the as-synthesized AgNWs; and (**d**) the corresponding HRTEM micrograph with FFT pattern (1) and obvious Ag lattice fringes pattern (2).

**Figure 4 materials-12-00401-f004:**
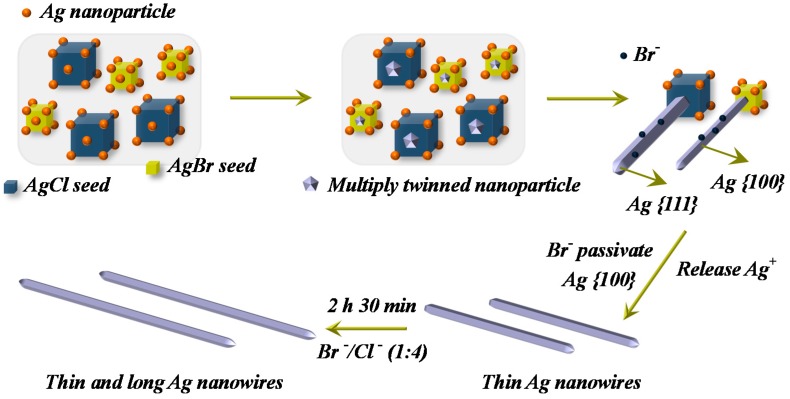
Schematic representation of the proposed formation mechanism of high aspect ratio AgNWs.

**Figure 5 materials-12-00401-f005:**
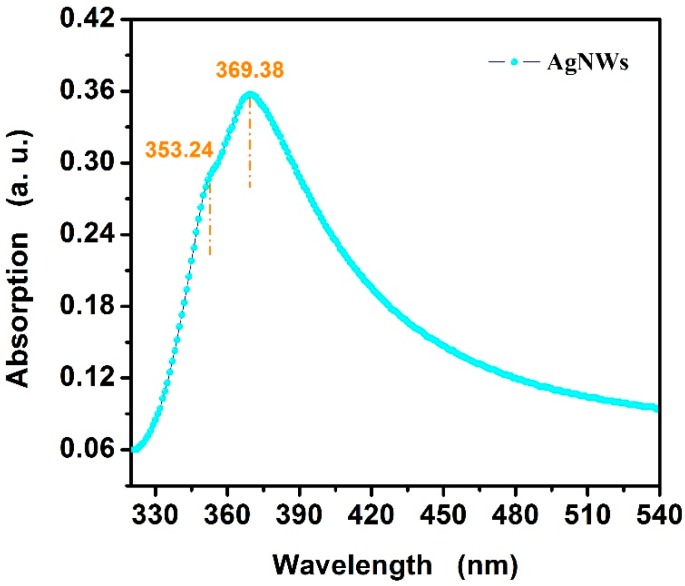
Typical UV-Vis absorption spectra of the as-synthesized AgNWs.

**Figure 6 materials-12-00401-f006:**
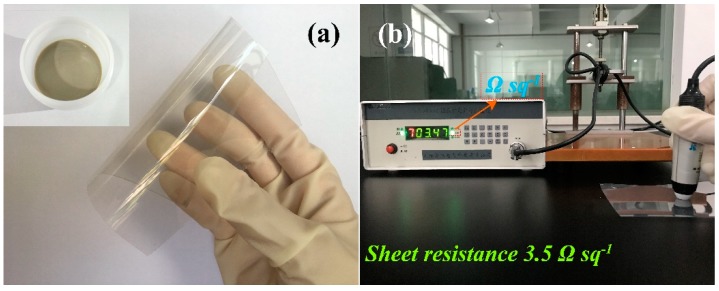
(**a**) Excellent flexible nature of the as-fabricated AgNW-based TCFs (the inset shows the photograph of AgNW-based conductive ink); and (**b**) the test picture of sheet resistance of the AgNW-based TCFs.

**Figure 7 materials-12-00401-f007:**
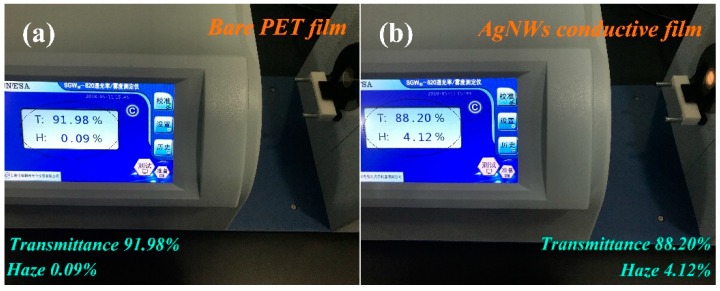
(**a**) The test picture of transmittance and haze based on bare PET substrate; and (**b**) a corresponding test picture based on AgNW-based TCFs.

**Figure 8 materials-12-00401-f008:**
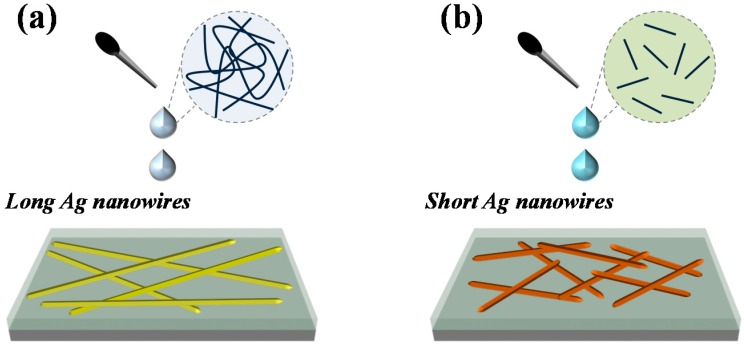
Schematic diagram of the mechanism for the ultra-low sheet resistance caused by ultra-long and high aspect ratio AgNWs. (**a**) Long Ag nanowires conductive network; (**b**) Short Ag nanowires conductive network.

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
