# Peer review of "One-Step Synthesis of Silver Nanowires with Ultra-Long Length and Thin Diameter to Make Flexible Transparent Conductive Films"

_materials, 2019, doi:10.3390/ma12030401_

Reviewer 1 Report

Dear authors,

Thank you for submitting your manuscript entitled “One-step synthesis of silver nanowires with ultra-long length and thin diameter to make flexible transparent conductive film with a low sheet resistance of 3.5 Ω“ to MDPI Materials. The synthesis of AgNWs has been extensively studied and the material is commercially already readily available. Thus, it is not easy to publish novel research on this subject. However, your study includes several new aspects such as i) the ratio of Br to Cl anions, ii) the use of HEC for the adhesion and iii) the fact that you really grow ultra-long wires using a one-step approach. In the context of using a one-step synthesis, you should also discuss the use of multistep growth protocols (such as here: DOI: 10.1021/cg301119d). I suggest to change your title to highlight the mean aspect ratio of 2500, which is a unique selling point, whereas the low resistance of 3.5 Ohm/sq. is not.

I recommend that your article will be published in MDPI Materials after major revisions. Please see my comments below:

1.      Title: I’m not a native speaker but you could think about changing the word make to produce and using the plural version for films.

2.      General remark: A native English speaker should screen the manuscript and place articles and use correct plural and singular, i.e. in the abstract it should be reaches (see page1, line16).

3.      Abstract: Most people (including me) don’t know the unit 0/FB for the adhesion. Can you include a statement whether that is good or not and discuss it in one line within the context that the adhesion of metal nanowire networks is in general a problem.

4.      Abstract: In addition to providing the minimum diameter and the maximum length, and respectively also the maximum aspect ratio, you should also provide the mean values in the abstract or at least the mean value for the aspect ratio (Page3, line 139 you state that these numbers are mean values already, then you should clarify it in the abstract).

5.      Abstract: Briefly explain the role of the bromine ions, e.g. co-passivating agent.

6.      Introduction: Page1, lines29-32, the sentence is incomplete, i.e. these materials were reported.

7.      Introduction: Page1, line34, please provide a reference.

8.      Introduction: Page1, line34, you talk about AgNWs and metal grids and in the next sentence, you write compared to AgNWs and metal grids, AgNWs are…..You should probably delete AgNWs in the first sentence, otherwise it makes no sense there.

9.      Introduction: Page2, line53, you should remove the full stop there.

10.   Introduction: The addition of halides to reduce the diameter of AgNWs is not new, please include a more profound discussion including the recent litertature: (DOI1: doi:10.1088/0957-4484/26/26/26520, DOI2: 10.1088/0957-4484/17/15/054, DOI3: 10.1021/acsnano.6b03806, DOI4: 10.1039/C5RA25310A and DOI5: 10.1155/2018/7304807).

11.   Introduction: The novelty of your work is that you’ve studied the ratio of Br to Cl and observed both a reduction in diameter, which is widely reported in the literature, and an increase in the nanowire length. Since these two parameters, the diameter and the length, are so important for your study, please include mean values considering a statistically significant number of nanowires. An automated way to extract the nanowire length is provided in this paper (DOI5: 10.1002/admi.201700568).

12.   2.1 Materials and methods: are chemicals no materials?

13.   Materials and methods: if this paper will be published by an MDPI journal, you have to include order number and companies more precisely for your products.

14.   Page3, line 93: You have introduced the abbreviation deionized already, please check this throughout the materials and methods section and also correct the typos.

15.   Results: Page3, lines 168-169: the twin morphology is not new. Please include references to other papers that show this fact.

16.   Results: Page6, line200: correct typo, i.e. range.

17.   Results: Page7, line222, please report the transmittance along with the sheet resistance.

18.   Results: Page7, line 224: discuss your result for the adhesion within a broader context, i.e. the adhesion is in general a problem and you found a way to address this.

19.   Results: Figure 6.b: You show the 4probe setup. Please include this measurement also in your materials and methods section and provide the probe current you have used.

20.   Results: Page8, lines239-246: Please include some references for papers that studied the dependence of the resistance on the aspect ratio of the wires.

Author Response

Dear reviewer,

We are truly grateful to yours critical comments and thoughtful suggestions. Based on these comments and suggestions, we have made careful modifications on the original manuscript. All changes made to the text are in yellow color. The point-by-point response has been uploaded by Word.  

Best wishes,

Yours sincerely,

Hongwei Yang

Reviewer 2 Report

     1. The English expression need to be improved. 

     2.You may explain more on the mechanism of decrease of AgNW diameter by KBr nucleant.

        You may show some data in comparision with NaBr/NaCl combination.

     3. "of 3.5 omega" in the title may be omitted.

Author Response

Dear reviewer,

We are truly grateful to yours critical comments and thoughtful suggestions. Based on these comments and suggestions, we have made careful modifications on the original manuscript. All changes made to the text are in blue color. The point-by-point response has been uploaded by Word.  

Best wishes,

Yours sincerely,

Hongwei Yang

Reviewer 3 Report

1# In Introduce part, line 34-37, some papers should be cited to prove the authors statement about "metal grids will cause serious moire interference phenomenon, and its production process need high costs."  Moreover, some recent papers (Sci. Rep. 2018, 8, 15167, and J. Mater. Chem. C 2018, 6, 7445-7461.) should be cited to prove "Compared to these alternative materials, AgNWs have high mechanical flexibility, excellent electrical conductivity and optical transparency." 

2# In Introduce part, line 61-63, the sentence "To our surprise, thus far, there has been virtually no reporting on a document that Br- decreased diameter of AgNWs while increasing length, and through controlled the growth of AgNWs to obtain AgNWs with a large aspect ratio." is not correct. In fact, several papers have reported synthesizing of AgNWs using Br- and Cl-, just like the author's synthesizing process (For example, Wiley group's work Nano Lett. 2015, 15, 6722-6766.) So this sentence can be revised as "To the best of our knowledge, only few papers have been reported that ....." and then cite Wiley group's work

3# In 2.2. Preparation of thin and long AgNWs, line 86-87,  "1:4 molar ratios of Br-/Cl- were dissolved into 160 mL EG". why the ratio is 1:4, not other ratio? Authors should give some explanation. 

4# In 2.4, can HEC be removed after heating at 130 degree for 5 min?

5# In Figure 2a, it can be clearly seen that the length of AgNWs has a wide distribution. Length distribution should be given.

6# In Figure 3a, the direction of the arrow is wrong. Moreover, why did the thickness of the PVP layer look like much thicker than that of AgNWs?

7# In Figure 7, for AgNW-TCFs' transmittance, did it include or exclude the transmittance of PET substrate?

Author Response

Dear reviewer,

We are truly grateful to yours critical comments and thoughtful suggestions. Based on these comments and suggestions, we have made careful modifications on the original manuscript. All changes made to the text are in green color. The point-by-point response has been uploaded by Word.  

Best wishes,

Yours sincerely,

Hongwei Yang

Round  2

Reviewer 1 Report

Dear authors,

Thank you for submitting your revised version. Since you've addressed all my comments of the 1th revision round very carefully, I would like to recommend the editorial board to accept your paper in the present form. I wish you good luck for your future research.

Reviewer 3 Report

The manuscript can be accepted in present form after minor revision of the some references.

1# Reference 18, the page number is not correct.

2# Reference 19 is a conference paper. The full name of this conference should be given or the format of the reference should meet the journal's requirement.